# FlexHiNM-GP: Flexible Hierarchical Pruning via Region Allocation and Channel Permutation

**Xiaodie Yi**[1], **Hayun Lee**[2], **Dongkun Shin**[1]*
[1]Department of Computer Science and Engineering, Sungkyunkwan University
[2]Department of Electrical and Computer Engineering, Sungkyunkwan University
`cindy.135luv@g.skku.edu, {lhy920806, dongkun}@skku.edu`

## Abstract

N:M sparsity has emerged as a hardware-friendly pruning strategy, notably supported by NVIDIA's Sparse Tensor Cores. While efficient, its fixed sparsity ratio restricts flexibility, making it difficult to adapt pruning granularity to varying weight importance across layers and architectures. To overcome this limitation, we propose FlexHiNM, a hybrid framework that adaptively partitions each layer into three regions: dense, vector-pruned, and N:M sparse, enabling finer-grained control while preserving hardware compatibility. To better preserve salient weights, we extend this to FlexHiNM-GP, which incorporates Gyro-Permutation, an iterative channel-rearrangement algorithm. Through successive sampling, clustering, and assignment, Gyro-Permutation aligns high-importance weights with structured sparsity patterns and mitigates suboptimal configurations in multi-level pruning. During gradual pruning, FlexHiNM-GP further employs a differentiable masking mechanism based on the Hard Concrete distribution, enabling gradient-based mask learning and preventing over-aggressive early pruning. Experiments on vision and language benchmarks demonstrate that FlexHiNM-GP consistently surpasses strong structured baselines and approaches the performance of unstructured pruning, validating the effectiveness of combining hybrid sparsity with learned masks and permutation strategies.

## 1 Introduction

Parameter pruning, zeroing out less important weights in a neural network, has become a key technique for managing the growing size of modern deep neural networks (DNNs). Pruning reduces both the number of floating-point operations (FLOPs) and the memory footprint required for model execution. A wide range of pruning strategies with varying granularity has been explored (Han et al., 2015; He et al., 2017; Tan et al., 2022; Li et al., 2016). In general, fine-grained methods such as unstructured pruning achieve minimal accuracy degradation at a given sparsity level. However, their highly irregular weight patterns lead to inefficient memory access, making it challenging to accelerate sparse model execution in practice.

To address this issue, N:M pruning (Mishra et al., 2021) has been proposed, which retains only the important $N$ elements in every group of $M$ elements within a vector, setting the rest to zero. N:M sparsity enables the removal of arbitrary elements within each group, similar to unstructured pruning, and thus helps preserve model accuracy after pruning. NVIDIA's Ampere architecture introduces Sparse Tensor Cores (STC) that support hardware-level sparse indexing for N:M operations, eliminating the need for additional software-level indexing overhead. A specialized instruction for sparse matrix multiplication-accumulation, `mma.sp`, performs the N:M sparse operation by accessing only the relevant input data as specified by the *NM index*.

However, N:M sparse pruning suffers from the limitation of fixed sparsity. Current hardware supports only specific N:M patterns (e.g. 2:4), resulting in a fixed 50% sparsity. To support variable

---

*Corresponding author.

sparsity levels, Venom (Castro et al., 2023) introduces a hierarchical compression technique that combines N:M sparsity with vector-wise sparsity. Specifically, several unimportant column vectors are first pruned entirely, and then row-wise N:M pruning is applied to the remaining vectors. By adjusting the column vector pruning ratio, the overall sparsity can exceed 50%, allowing for more flexible sparsity control. This composite sparsity scheme is referred to as **hierarchical N:M (HiNM) sparsity**.

When executing HiNM-pruned models, GPU kernel code uses a *vector index* of the remaining vectors to load only the corresponding inputs from global memory into shared memory. Subsequently, N:M sparse operations are performed using the associated NM index.

As illustrated in Figure 1a, HiNM such as Venom (Castro et al., 2023) partitions column vectors based on their importance scores ($R$), defined as the fraction of significant elements within each column. Vectors with low scores are entirely pruned (i.e., 0:4 pruning), while the rest undergo 2:4 pruning. However, the wide variance in importance scores among retained vectors may reduce the effectiveness of uniform 2:4 pruning. For instance, even if all four elements in a vector are important, the 2:4 constraint requires that two of them be pruned, potentially discarding useful information.

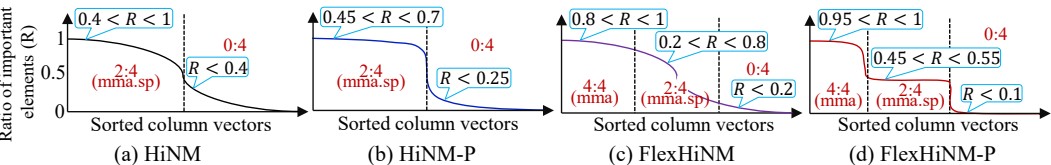

Figure 1: HiNM variants

To address the score variance problem, we consider two complementary approaches. The first approach is channel permutation (Pool & Yu, 2021), which rearranges the order of column vectors to more uniformly distribute important elements across row vectors. As illustrated in Figure 1b, the permuted variant, termed **HiNM-P**, reduces the range of importance scores among vectors subject to N:M pruning. However, channel permutation alone cannot perfectly balance the score distribution across all vectors.

The second approach introduces a three-level pruning scheme. Instead of uniformly applying 2:4 pruning to all vectors retained after vector-level pruning, the retained vectors are divided into two regions: a dense region and a 2:4 sparse region (Figure 1c). This scheme, called **FlexHiNM** (Flexible HiNM), preserves entire vectors with high importance scores, while vectors with mid-range scores are pruned under the 2:4 sparsity constraint. During execution, dense vectors are processed using standard matrix multiply-accumulate instructions (`mma`), while 2:4 sparse vectors are executed using specialized sparse instructions (`mma.sp`). A key challenge in FlexHiNM is determining the region boundaries (i.e., the proportions of 4:4, 2:4, and 0:4 regions) for a given target sparsity.

While the three-level sparsity scheme allows vectors to adopt different sparsity levels, it does not ensure that the 2:4 sparse region conforms well to the rigid pruning constraint. To address this, Flex-HiNM is extended with channel permutation, forming **FlexHiNM-P** (Figure 1d). This arrangement guarantees that dense-region vectors consist entirely of high-score elements, while in the 2:4 sparse region roughly half of the elements in each vector are high-score. Combining three-level sparsity with channel permutation is thus critical for fine-grained control and structural compatibility with 2:4 pruning. To realize this, we propose *gyro-permutation* (GP), a FlexHiNM-aware channel permutation method tailored to the HiNM sparsity structure.

The FlexHiNM scheme enhanced with GP, referred to as **FlexHiNM-GP**, integrates four key components:

1. **Region boundary search** for three-level sparsity: Given a target sparsity level, this algorithm optimizes the proportion of each region (4:4, 2:4, and 0:4) to maximize the total importance score of the preserved weights.

2. **Gyro-permutation** technique: A channel reordering method that jointly optimizes the order of input and output channels to better align with both vector-level and 2:4 sparsity constraints.

3. **Gradual pruning and fine-tuning flow:** We propose a gradual pruning framework for HiNM-GP in which weight updates and 2:4 mask learning co-evolve during stepwise fine-tuning. As sparsity increases, both the weights and the pruning boundaries change, rendering fixed heuristic selections suboptimal after fine-tuning. To address this, we jointly optimize the 2:4 masks and the weights using the Hard Concrete distribution, enabling the masks to adapt to evolving weight importance while strictly preserving structural constraints.

4. **Custom GPU kernel for FlexHiNM-GP**: We design an optimized GPU kernel that dynamically reorders input channels at runtime to ensure cross-layer consistency, thereby eliminating the need for static offline permutation. The kernel processes the 2:4 sparse region and the denser region using separate specialized kernels, and subsequently merges their outputs to produce the final result.

Experimental results on various DNN models including Deit, Bert, and Llama2, demonstrate that FlexHiNM-GP significantly improves accuracy, achieving performance comparable to that of unstructured sparsity.

## 2 RELATED WORK

**Weight pruning** Numerous pruning strategies have been developed, incorporating various sparsity patterns (He et al., 2017; Han et al., 2016; Li et al., 2016), salience estimation methods (Sanh et al., 2020; Frantar et al., 2021; Kurtic et al., 2022), and fine-tuning approaches such as one-shot (Lee et al., 2019) and gradual pruning (Kurtic et al., 2022). Irregular fine-grained sparsity often minimizes accuracy loss but increases indexing overhead. Vector-wise sparsity (Huang et al., 2022; Tan et al., 2022; Park et al., 2023), which prunes vectors of shape $V \times 1$, provides a better balance between accuracy and efficiency. N:M sparsity has also been advanced with training methods (Zhou et al., 2021; Hubara et al., 2021), layer-wise allocation assuming variable N:M support (Sun et al., 2021), and optimized GPU kernels (Sun et al., 2022). Hierarchical sparsity has been explored in both hardware (Wu et al., 2023; Liu et al., 2023) and software. Venom (Castro et al., 2023) combines vector-wise sparsity handled in software with N:M sparsity supported by hardware.

**Channel permutations** Channel permutation techniques (Huang et al., 2022; Tan et al., 2022; Pool & Yu, 2021; Ji et al., 2018) reorder weight matrices by adjusting input or output channels. This aligns weights with the required sparsity pattern and enables efficient removal of unimportant elements. Prior works mainly target single-level sparsity: for column-wise vector sparsity, output channel permutation groups unimportant vectors for removal (Tan et al., 2022); for N:M sparsity, input channel permutation redistributes significant elements across rows (Pool & Yu, 2021). Tetris (Ji et al., 2018) extends this to block sparsity (e.g., $4 \times 4$) by swapping channels in both dimensions and introducing index translation across layers, though at high GPU overhead. Our Gyro-Permutation instead integrates index translation into the native indexing of HiNM sparsity during memory transfer, eliminating extra runtime cost.

**Learnable semi-structured sparsity** Recent works have also explored learnable sparsity, where pruning patterns are optimized jointly with training. MaskLLM (Fang et al., 2024) formulates N:M sparsity as a discrete distribution via Gumbel Softmax, enabling end-to-end mask learning but suffering from scalability and costly sampling. S2HPruner (Lin et al., 2024) introduces a soft-to-hard mask distillation framework that couples a continuous mask with its discretized version. However, its stability and final patterns depend heavily on hyperparameter tuning and heuristic schedules, limiting generalization across architectures and sparsity ratios.

## 3 FLEXHINM SPARSE SYSTEM

### 3.1 MULTI-LEVEL PRUNING

FlexHiNM sparsity pruning adopts a three-level pruning strategy that applies vector-wise pruning and N:M structured pruning to different regions of the weight tensor. As illustrated in Figure 2, the original weight matrix is first partitioned into multiple tiles, each corresponding to a group of column vectors associated with the same output channel index. The height of each tile is determined

by the vector size used for vector-level pruning. In FlexHiNM-GP, output channel permutation is applied across tiles, while input channel permutation is performed independently within each tile.

Within each tile, vectors with extremely low importance scores are fully pruned using outer-vector-wise pruning (OVW) (Tan et al., 2022). Among the remaining vectors, those with high importance scores are selected and grouped into a separate dense matrix (shown in red), with their corresponding vector indices recorded. It is important to note that the number of selected vectors must be consistent across all tiles to maintain structural alignment during execution. The remaining vectors (shown in blue) are pruned according to the 2:4 sparsity pattern. These pruned vectors are stored in a compact sparse format containing only non-zero elements, along with the corresponding NM indices.

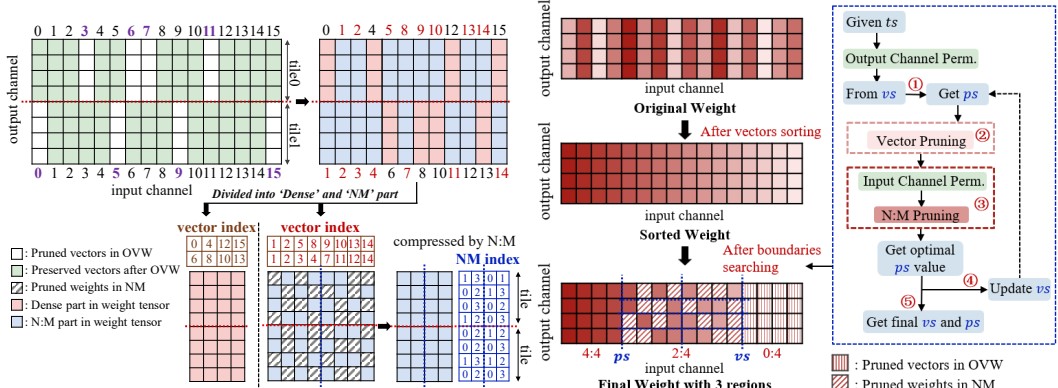

Figure 2: Pruning pipeline    Figure 3: Adaptive region-wise sparsity allocation

## 3.2 BOUNDARY SEARCH

Our method introduces a unified framework that jointly allocates structured sparsity across vector and element levels. As illustrated in Figure 3, the weight matrix is partitioned into three disjoint regions: **Dense region (4:4)**, **N:M sparse region (2:4)**, and **Pruned region (0:4)**.

To achieve a target sparsity $ts$, we derive a closed-form relationship between two boundary parameters: the *vector sparsity boundary* ($vs$) and the *partial sparsity boundary* ($ps$). Here, $vs$ denotes the fraction of fully pruned vectors, while $ps$ represents the fraction of the remaining vectors assigned to N:M sparsity after vector pruning.

Consider a weight matrix comprising $K$ column vectors. Vector pruning leaves a fraction $(1 - vs)$ of vectors intact. Of these remaining vectors, a fraction $ps$ is further subjected to the 2:4 structured sparsity constraint, which retains 50% of their elements. Consequently, the sparsity contributed by 2:4 pruning is $(1 - vs)\, ps \times 0.5$. Combining this with vector pruning ($vs$) yields the total sparsity $ts = vs + (1 - vs)\, ps \times 0.5$. Rearranging, we obtain:

$$ps = \frac{2(ts - vs)}{1 - vs} \tag{1}$$

For example, in Figure 2 with $ts = 50\%$ and $vs = 25\%$, we compute $ps = 66.7\%$. This means 4 out of 16 vectors (25%) are fully pruned (i.e., channels 3, 6, 7, and 11 in tile0), 8 vectors ($12 \times 66.7\%$) are placed in the 2:4 sparse region (i.e., channels 1, 2, 5, 8, 9, 10, 13, and 14), and the remaining 4 vectors (i.e., 0, 4, 12, and 15) are retained for dense computation.

The feasible space of Equation 1 is shown in Figure 4, where each curve depicts the trade-off between $vs$ and $ps$ for a fixed target sparsity $ts$. To select the optimal boundary pair for a given $ts$, we employ an iterative search strategy, illustrated as steps ①–⑤ in Figure 3 (right blue box). The steps are described below, with the overall pruning process summarized in Algorithm 1.

① **Output Channel Permutation & Boundary Initialization:** Prior to vector-level pruning, we apply an output channel permutation using Gyro-Permutation to enhance pruning effectiveness, corresponding to the function GyroPerm(weights,axis=output) in Step 1 of Algorithm 1. For

a target $ts$, we begin with $ps = 0$, meaning no N:M sparse region. For instance, when $ts = 50\%$ (red curve in Figure 4), the initial values are $vs = 0.5$ and $ps = 0$.

② **Vector Pruning:** The column vectors are sorted in descending order of importance to identify those with minimal impact on model capacity. Column-vector pruning is subsequently performed via `VectorPrune(weights, vs)` in Step 4 of Algorithm 1.

---

**Algorithm 1** FlexHiNM Sparse Pruning Pipeline

---

**Require:** Weight matrix $W$, target sparsity $ts$, vector size $v$
**Ensure:** Updated weight $W'$
1:  Split $W$ into tiles $\{W^t\}$ of size $v$, then $\{W^t\} \leftarrow \text{GyroPerm}(W, \text{axis=output})$;
2:  **for** each $W^t$ **do**
3:      **for** $vs \leftarrow ts$ to 0 step $-\alpha$ **do**
4:          Compute $ps$ using Equation 1; $W^t \leftarrow \text{VectorPrune}(W^t, vs)$;
5:          $W^t \leftarrow \text{NMPrune}(\text{GyroPerm}(W^t, \text{axis=input}), ps)$;
6:          Evaluate $R_{\text{total}} = R_{\text{dense}} + R_{24}$ ;
7:          **if** $R_{\text{total}} > R_{\text{best}}$ **then** $vs_{best} \leftarrow vs$; $ps_{best} \leftarrow ps$; $R_{\text{best}} \leftarrow R_{\text{total}}$;
8:          **else break**
9:          **end if**
10:     **end for**
11: **end for**
12: $W_{\text{pruned}} \leftarrow \text{GetVWPruned}(\text{VectorPrune}(W, vs_{\text{best}}))$;
13: $W_{\text{dense}} \leftarrow \text{GetDense}(W \setminus W_{\text{pruned}}, ps_{\text{best}})$;
14: $W_{2:4} \leftarrow \text{NMPrune}(W \setminus (W_{\text{dense}} \cup W_{\text{pruned}}), ps_{\text{best}})$ ;
15: Return $W' = W_{\text{dense}} \cup W_{2:4} \cup W_{\text{pruned}}$

---

③ **N:M Pruning:** Among the remaining weight vectors, a subset is designated for N:M sparsity. Specifically, the less important column vectors (a fraction $ps$ of the vectors) undergo row-wise 2:4 pruning, generating the corresponding N:M indices. Since N:M pruning follows a row-wise 2:4 pattern, we apply an input-channel permutation via `GyroPerm(weights, axis=input)` before performing pruning. It is important to note that the input channel dimension must be divisible by 4 to satisfy the hardware constraints of NVIDIA's Sparse Tensor Core architecture.

④–⑤ **Repetition or Termination:** To guide the search for the optimal $(vs, ps)$ pair, we maximize $R_{\text{total}} = R_{\text{dense}} + R_{24}$, where $R_{\text{dense}}$ and $R_{24}$ denote the cumulative importance scores of the dense and N:M sparse regions, respectively, computed using the second-order importance metric (Kurtic et al., 2022).

Rather than exhaustively exploring all $(vs, ps)$ pairs, we exploit the concavity of the objective (proof in Appendix B). The search proceeds by decreasing $vs$ with step size $\alpha$ and terminates when the objective value begins to decrease. To prevent overly aggressive decline, the step size is updated as $\alpha = \alpha - \beta vs - \gamma \text{RMSProp}(vs, ps)$, where $\beta$ penalizes excessively large $vs$ values and $\gamma$ adjusts the soft constraint via the RMSProp term. The configuration at the last monotonic improvement is chosen as the final solution, and pruning is then executed according to the resulting boundary values.

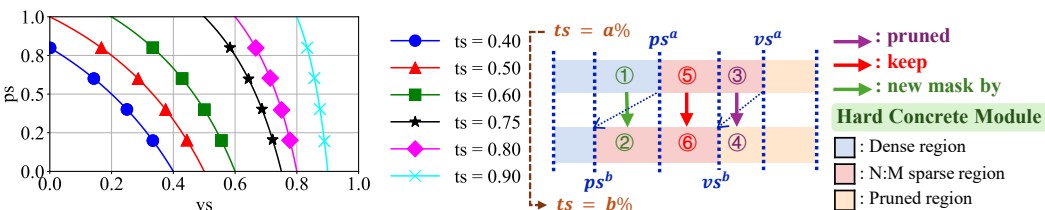

Figure 4: Relationship between $vs$ and $ps$

Figure 5: Mask reconfiguration between sparsity levels

## 3.3 FLEXHINM KERNEL

Figure C.1 in Appendix illustrates the kernel design for executing models pruned with FlexHiNM-GP. As illustrated in Figure 2, the weight tensor is separated into sparse and dense tiles, each assigned to a different CUDA stream: **Stream0** for sparse tiles and **Stream1** for dense tiles. It is crucial to note that different tiles use different vector indices. Initially (①), both the weight and input matrices are stored in global memory. These matrices are transferred to the shared memory of each Stream Multiprocessor (②③), and subsequently loaded into the registers of the processing cores (④⑤). To fully utilize GPU parallelism, multiple thread blocks across different SMs process tiles concurrently. Additionally, each thread block loads the relevant rows of the input matrix into shared memory, specifically the input row vectors that correspond to the column vector indices of the weight tile.

**Stream0: Sparse Tile Execution** Each thread block loads its pruned weight tile and only relevant input vectors (e.g. indices 1, 2, 5, 8, etc.) into shared memory. For each output row, the STC engine fetches pruned weights and corresponding inputs via precomputed N:M indices (e.g. 1,3; 0,1) and performs `mma.sp` in registers, minimizing memory access and enhancing throughput.

**Stream1: Dense Tile Execution** In parallel, dense tiles (e.g. thread blocks 2 and 3) load full tiles with all vector indices (e.g. 0, 4, 12, 15) and use standard `mma` for dense GEMM computation via tensor cores.

**Final Synchronization** Results from Stream0 and Stream1 are written directly into the global output matrix, where their partial contributions are accumulated using `atomicAdd` without requiring inter-stream synchronization. The complete kernel workflow and execution timeline are illustrated in Appendix C.

## 4 MASK LEARNING

### 4.1 GRADUAL PRUNING WORKFLOW

We adopt gradual pruning, progressively increasing sparsity to avoid abrupt accuracy degradation, similar to oBERT (Magic, 2023). At each stage, both pruning boundaries and the set of weights assigned to the 2:4 region may change. A static greedy selection becomes suboptimal once weights are updated through fine-tuning. Therefore, we jointly optimize both remaining weights and 2:4 masks during each pruning stage. We parameterize masks using the Hard Concrete distribution, enabling differentiable optimization under strict structural constraints , which allows masks to adapt to evolving weight importance while maintaining monotonic sparsity progression.

As shown in Figure 5, when the target sparsity increases from $a\%$ to $b\%$ during gradual pruning, the pruning boundaries are updated accordingly. Channel permutation is performed only during the boundary search stage and is kept fixed throughout training. As a result, some vectors in the previous N:M sparse region become fully pruned (③ → ④), while new vectors transition from the dense region to the N:M sparse region (① → ②), where fresh masks are generated using the Hard Concrete module. This design ensures monotonic and irreversible pruning (see Appendix D for proof) by restricting mask updates to newly introduced weights only.

### 4.2 HARD CONCRETE MODULE

To enable differentiable mask learning under the N:M sparsity constraint, we adopt the Hard Concrete distribution (Maddison et al., 2016) to parameterize binary masks in a continuous and learnable form.

**Step.1 Mask sampling** : Each weight $w_i$ within a 4-element group is assigned a learnable logit $\alpha_i$ and paired with uniform noise $\epsilon_i \sim \mathcal{U}(0,1)$ to generate soft masks $z_i$, from which exactly two elements are retained per group.

**Step.2 Generate soft mask** $z_i$ : Given the sampled noise and logits from Step 1, we compute an intermediate variable $s_i$ that serves as a temperature-scaled activation.

$$s_i = \sigma\left(\frac{1}{\tau}(\log \epsilon_i - \log(1 - \epsilon_i) + \log \alpha_i)\right) \qquad (2)$$

where $\sigma(\cdot)$ is the sigmoid function and $\tau \in [0.1, 1.0]$ is the annealed temperature. To enable mask learning over a broader range, $s_i$ is first stretched to the range $(\gamma, \zeta) = (-0.1, 1.1)$, and then clipped to $[0, 1]$ to obtain the final soft mask $z_i$: $\bar{s}_i = s_i \cdot (\zeta - \gamma) + \gamma$, $z_i = \min(1, \max(0, \bar{s}_i))$.

Then $z_i$ is applied to the corresponding weights through element-wise multiplication during forward propagation $\bar{w}_i = w_i \cdot z_i$. Both the model weights and the sampling logits $\alpha_i$ are updated via backpropagation, guided by a composite loss:

$$L = L_{\text{task}} + L_{\text{sparse}} + L_{\text{hard}}, \quad L_{\text{sparse}} = \lambda_s \, \text{mean}(z_i), \quad L_{\text{hard}} = \lambda_c \, \text{mean}\left( \left| \sum_{i=1}^{4} z_i - 2 \right| \right) \quad (3)$$

where $L_{\text{task}}$ denotes the standard task loss, $L_{\text{sparse}}$ encourages overall sparsity by reducing the average mask value, and $L_{\text{hard}}$ enforces strict 2:4 sparsity within each group to avoid invalid pruning patterns; $\lambda_s$ and $\lambda_c$ control the relative strengths of these two regularization terms, balancing global sparsity against strict pattern validity during training. Implementation details are provided in Appendix E.

**Step.3 Hardening and binary mask generation** : During training, the temperature $\tau$ is gradually annealed every 5 epochs, making the soft mask to hard mask.

$$z_i = \begin{cases} 1, & \text{if } z_i > 0.5 \\ 0, & \text{otherwise} \end{cases} \quad (4)$$

Finally, at the end of each 20-epoch interval, we convert the soft mask into a hard binary mask by applying a 0.5 threshold to $z_i$, as defined in Equation 4.

## 5 GYRO-PERMUTATION

Channel permutation aligns salient weights with structured sparsity patterns by rearranging channels. Given a pre-trained model and target sparsity, pruning applies a binary mask $M$ to each layer's weight matrix $W \in \mathbb{R}^{m \times n}$, retaining important weights while enforcing structural constraints ($M_{ij} = 0$ indicates pruning). We assume an importance score matrix $D \in \mathbb{R}^{m \times n}$ aligned with $W$. The pruning mask $M$ must satisfy three constraints: (1) $C_v$, enforcing binary pruning at the column-vector level; (2) $C_{2:4}$, restricting each four-aligned row vector to two nonzeros; and (3) $C_s$, meeting the global target sparsity. The channel permutation problem can thus be formulated as the following optimization:

$$\underset{\Lambda_O, \Lambda_I^0, \dots, \Lambda_I^{T-1}}{\arg\max} \quad \|M \odot D\,[\Lambda_O; \Lambda_I]\| \quad \text{s.t.} \quad M \text{ satisfies } C_v, C_{2:4}, \text{ and } C_s, \quad (5)$$

Here, $\Lambda_O$ and $\Lambda_I^k$ denote the output channel order and input permutations for the $k$-th tile, respectively. The notation $D[\Lambda_O; \Lambda_I]$ indicates the reordered score matrix, and $\|\cdot\|$ denotes the L1 norm.

We decompose Equation 5 into two sub-problems: output channel permutation and tile-wise input channel permutation. Accordingly, the Gyro-Permutation framework follows a pipeline of output permutation, column-vector pruning, and input permutation. Each iteration comprises three stages: sampling, clustering, and assignment, as illustrated in Figure 6, and can be repeated to progressively refine the permutation by capturing interactions between output and input structures.

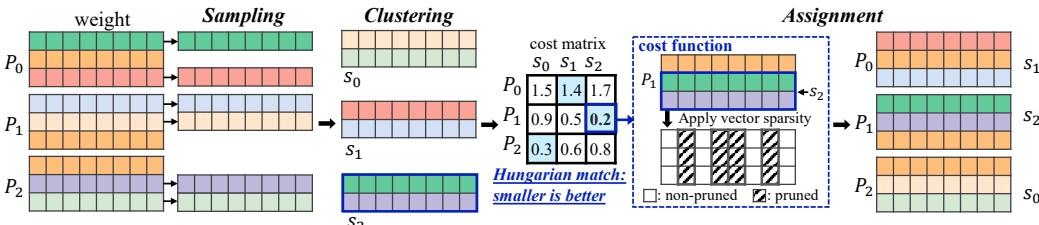

Figure 6: Gyro-Permutation flow

The original weight matrix is partitioned into $N$ tiles ($P_0, P_1, P_2$), each with a height equal to the vector size. From each tile, $M$ vectors are **sampled** and then grouped into $N$ clusters ($s_0, s_1, s_2$)

through balanced K-means **clustering**, with each cluster containing $M$ vectors. Pruning costs $C_{i,j}$ are computed for **assigning** cluster $s_j$ to tile $P_i$ (e.g., $C_{1,2} = 0.2$ for assigning $s_2$ to $P_1$), and the optimal assignment is obtained via the Hungarian algorithm. Implementation details are provided in Appendix F.

## 6 EXPERIMENTS

We conduct experiments on three families of models, Deit, Bert, and LLaMA-2. For image classification, we evaluate Deit-Small and Deit-Base on ImageNet-1K using vector sizes of 64 and 128. For NLP benchmarks, we test Bert-Base on QQP, SST-2, and SQuAD v1.1, and LLaMA-2 7B on six 5-shot reasoning datasets. Each model is pruned at target sparsity levels of 75% to 95%, and evaluated under both vector sizes where applicable.

All experiments are run on NVIDIA RTX 4090 GPU with gradual pruning schedule starting from 30% sparsity, increasing to the target in four stages. Each stage begins with mask reinitialization, followed by 20 epochs of joint optimization of weights and Hard Concrete masks.

### 6.1 ACCURACY EVALUATION

Figure 7 presents the results for transformer models using vector-size of 64: `Unstructured` corresponds to element-wise pruning; `HiNM-GP` represents the HiNM pruning with Gyro-permutation; `FlexHiNM-GP` denotes the flexible Hierarchical pruning via region allocation and Gyro-permutation; `OVW` denotes the traditional out-vector-wise pruning (Tan et al., 2022), as a baseline since it can be regarded as a special case of our framework, representing a lower bound of accuracy for FlexHiNM as illustrated in Figure 9; `HiNM-V` is a variant of our method that omits the Gyro-Permutation process, which is mainly equivalent to Venom, to isolate the effect of channel permutation.

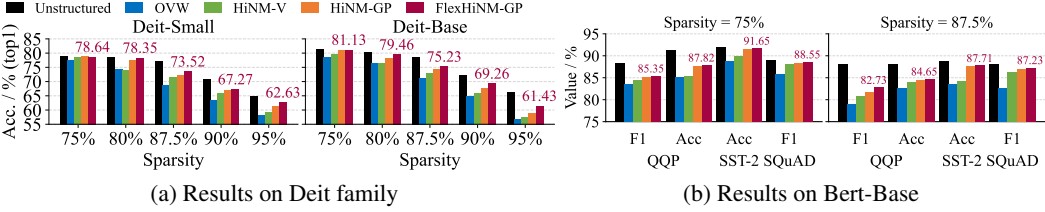

(a) Results on Deit family      (b) Results on Bert-Base

Figure 7: Gradual pruning for transformer models (V=64)

`FlexHiNM-GP` achieves the highest top-1 accuracy among structured pruning techniques on Deit-Base under moderate sparsity, reaching 81.13% and 79.46% at 75% and 80% sparsity, respectively (Figure 7a). Comparing `HiNM-V` and `HiNM-GP` highlights the role of channel permutation: without it, `HiNM-V` suffers from misalignment between salient weights and preserved tiles, causing greater degradation as sparsity increases; `HiNM-GP` alleviates this by redistributing important elements across row groups through gyro-permutation, improving compatibility with the 2:4 constraint, while `FlexHiNM-GP` further improves performance by coupling permutation with flexible region allocation.

As shown in Figure 7b, `FlexHiNM-GP` achieves the best structured pruning performance on Bert, attaining 88.55 F1 on SQuAD and 91.65% accuracy on SST-2 at 75% sparsity, closely matching the unstructured baseline (89.04 and 91.86%, respectively). At 87.5% sparsity, where all methods exhibit noticeable degradation, `FlexHiNM-GP` remains the most robust, reaching 87.23 on SQuAD and 87.71% on SST-2 and consistently outperforming `HiNM-GP` by small margins.

Table 1 reports the performance of different pruning strategies on six downstream benchmarks on LLaMA2-7B at sparsity levels of 75% and 87.5%. Unstructured pruning yields the highest raw scores but suffers from large task variance and lacks hardware support, while structured methods such as `OVW` and `HiNM-V` incur substantial degradation. Incorporating channel permutation in `HiNM-GP` alleviates part of the drop, and `FlexHiNM-GP` further improves average accuracy by +1.39% over `HiNM-GP` at 75% sparsity, with consistent gains across both reasoning-oriented

and commonsense tasks. When sparsity increases to 87.5%, all methods decline preserve the same ranking, and `FlexHiNM-GP` consistently outperforms `HiNM-GP`.

| Method | 75% | | | | | | 87.5% | | | | | |
|---|---|---|---|---|---|---|---|---|---|---|---|---|
| | OBQA | ARC-C | ARC-E | PIQA | BoolQ | HellaS | OBQA | ARC-C | ARC-E | PIQA | BoolQ | HellaS |
| Dense | 32.07 | 43.08 | 76.39 | 79.62 | 77.74 | 57.03 | 32.07 | 43.08 | 76.39 | 79.62 | 77.74 | 57.03 |
| Unstructured | 23.87 | 33.26 | 62.04 | 71.75 | 59.13 | 45.07 | 20.53 | 29.12 | 57.63 | 67.84 | 54.02 | 40.96 |
| OVW | 19.67 | 27.94 | 53.77 | 65.13 | 52.27 | 39.86 | 15.87 | 23.75 | 48.92 | 61.32 | 48.56 | 34.28 |
| HiNM-V | 20.33 | 28.58 | 54.96 | 66.16 | 53.18 | 40.63 | 16.73 | 24.56 | 50.34 | 62.45 | 49.61 | 35.17 |
| HiNM-GP | 22.07 | 30.17 | 57.84 | 68.87 | 55.91 | 42.72 | 18.07 | 26.33 | 52.71 | 64.18 | 51.37 | 36.82 |
| **FlexHiNM-GP** | **23.13** | **31.62** | **59.22** | **70.14** | **57.23** | **44.58** | **19.20** | **27.58** | **54.38** | **65.72** | **52.46** | **38.11** |

Table 1: Llama2-7B Downstream task performance (%)

Consequently, these results indicate that `FlexHiNM-GP` narrows the gap to unstructured pruning while retaining full hardware compatibility, achieving a more favorable trade-off between accuracy and structured sparsity, stabilizing performance under high compression while preserving task generalization in large-scale language models. Additional evaluations under extended sparsity levels (up to 95%) and different vector sizes (64 and 128) are reported in Appendix G to further validate robustness across model scales and compression regimes.

## 6.2 INFERENCE EVALUATION

We evaluate the inference efficiency of the proposed sparsity schemes using our custom Tensor Core kernel on NVIDIA RTX 4090 GPU. The benchmark is conducted on Deit-Base and Bert-Base with batch size set to 1, averaged over 30 post–warm-up runs.

The observed latency behavior is consistent with the memory-access characteristics of each pruning rule, as shown in Figure 8. `OVW` achieves the fastest execution, as it loads only the input elements required for computation. `HiNM-GP`, in contrast, loads full input vectors but uses only a subset of elements determined by the 2:4 pattern, resulting in additional unused loads that reduce efficiency relative to `OVW`.

`FlexHiNM-GP` processes both dense (4:4) vectors and 2:4-pruned vectors, resulting in the compute pattern that lies between `OVW` and `HiNM-GP`, with latency naturally interpolating between the two baselines. This trend is consistent across both Deit-Base and Bert-Base. Notably, `FlexHiNM-GP` maintains these inference gains while preserving partial dense regions, which contributes to its higher accuracy. The small latency overhead introduced by these dense vectors is thus well justified when considering the trade-off of accuracy-efficiency.

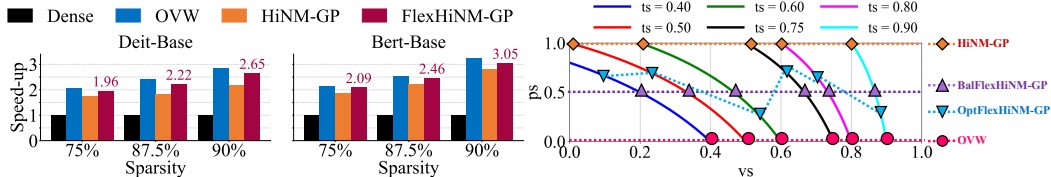

Figure 8: Speed comparison across sparsity levels      Figure 9: Variants based on boundaries

## 6.3 ABLATION STUDY

### 6.3.1 COMPARISON WITH GUMBEL SOFTMAX

Table 2 illustrates the ablation study over six variants to compare different noise functions used for mask learning. Variant ① is our proposed `FlexHiNM-GP` which achieves the highest overall average accuracy among all variants, while Variant ⑥ corresponds to the `HiNM-GP` baseline. Between them we construct four intermediate variants by toggling the use of Flexible Distribution (F), and switching between Hard Concrete (H) and Gumbel-Softmax (G) for mask generation.

While Gumbel-Softmax enables differentiable 2:4 pruning by selecting from six fixed patterns per group (Fang et al., 2024), its limited search space restricts fine-grained saliency modeling. In contrast, Hard Concrete generates continuous, near-binary masks independently for each element, offering greater flexibility while still supporting structured enforcement. It also facilitates stable gradient

flow through hard-sigmoid activation and avoids categorical sampling, making it a more scalable choice for structured sparsity learning.

| V | F | H | G | 75% | 80% | 87.5% | 90% | 95% |
|---|---|---|---|-----|-----|-------|-----|-----|
| ① | ✓ | ✓ | – | **81.13** | **79.46** | **75.23** | **69.26** | **61.43** |
| ② | ✓ | – | ✓ | 81.10 | 79.27 | 75.55 | 68.66 | 59.77 |
| ③ | ✓ | – | – | 81.08 | 79.12 | 75.34 | 68.50 | 59.64 |
| ④ | – | ✓ | – | 81.07 | 78.49 | 74.68 | 68.12 | 59.19 |
| ⑤ | – | – | ✓ | 80.82 | 78.77 | 74.91 | 68.29 | 59.42 |
| ⑥ | – | – | – | 81.04 | 78.28 | 74.35 | 67.64 | 58.94 |

Table 2: Ablation results under variants on Deit-Base

| Variant | 75% | 80% | 87.5% | 90% | 95% |
|---------|-----|-----|-------|-----|-----|
| HiNM-GP | 81.04 | 78.28 | 74.35 | 67.64 | 58.94 |
| OVW | 78.67 | 76.34 | 71.26 | 64.87 | 56.82 |
| BalFlexHiNM | 80.38 | 78.51 | 75.09 | 68.45 | 57.26 |
| **OptFlexHiNM** | **81.13** | **79.46** | **75.23** | **69.26** | **61.43** |

Table 3: Ablation results under boundaries on Deit-Base

### 6.3.2 COMPARISON WITH BALANCED FLEXHINM-GP

Figure 9 compares four variants under different boundary settings defined by $ps$ and $vs$. `OVW` (pink) applies only vector pruning as $ps = 0$; `HiNM-GP` (orange) uses only N:M sparse on the whole dense part after `OVW` since $ps = 1$; `BalFlexHiNM-GP` (Balanced FlexHiNM, purple) fixes at $ps = 0.5$; and `OptFlexHiNM-GP` (Optimal FlexHiNM, blue), representing our core method, adaptively searches both boundaries to achieve the best performance.

Table 3 demonstrated that `OptFlexHiNM` consistently achieves the highest accuracy across all sparsity levels, which outperforms `HiNM-GP` by +1.62% at 90% sparsity and +2.49% at 95% sparsity, indicating that greedy boundary optimization effectively preserves important information while enforcing structured sparsity. Unlike manually fixed strategies such as `OVW` or `BalFlexHiNM`, the adaptive approach delivers a better trade-off between accuracy and structural regularity, which is particularly crucial for ViT models sensitive to pruning design.

### 6.3.3 EFFECT OF GYRO-PERMUTATION

Finally, we provide a direct ablation comparing `FlexHiNM` (`FlexHiNM-GP` without Gyro-Permutation) with `FlexHiNM-GP` at 75% sparsity (vector size = 64), isolating the effect of Gyro-Permutation. As shown in Table 4, GP consistently improves performance across all Bert-Base tasks; for example, `FlexHiNM-GP` improves QQP F1 from 84.78 to 85.35 and SQuAD F1 from 88.04 to 88.55.

Table 4: Impact of Gyro-Permutation on FlexHiNM

| Method | QQP | | SST-2 | SQuAD |
|--------|-----|-----|-------|-------|
| | F1 | Acc | Acc | F1 |
| FlexHiNM-GP | 85.35 | 87.82 | 91.65 | 88.55 |
| FlexHiNM | 84.78 | 86.42 | 90.60 | 88.04 |

The accuracy gaps between `FlexHiNM` and `FlexHiNM-GP` show that three-region partitioning alone does not fully resolve channel misalignment introduced by structured N:M sparsity. Gyro-Permutation provides the additional alignment needed for stable improvements, and its effect persists across tasks, even at moderate pruning ratios. These results confirm that the benefit of GP is not setting-specific, but stems from correcting the structural mismatch remaining after hierarchical partitioning.

## 7 CONCLUSION

We present FlexHiNM-GP, a unified framework for multi-level sparsity that combines adaptive region partitioning with permutation-aware pruning. It learns dynamic boundaries for column-wise and N:M sparsity, while Gyro-Permutation efficiently aligns salient weights through coordinated channel permutation. With Hard Concrete-based differentiable mask learning, FlexHiNM-GP achieves higher accuracy across diverse sparsity levels, demonstrating its effectiveness for large-scale model compression.

ACKNOWLEDGMENTS

This work was supported by Samsung Research Funding & Incubation Center of Samsung Electronics under Project Number SRFC-IT2402-15.

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

APPENDIX

## A    THE USE OF LARGE LANGUAGE MODELS (LLMS)

No but not all. We only used LLMs to polish some sentences.

## B    CONCAVE OPTIMAL TARGET

We consider the remaining weight magnitude $R$ as a function of vector-wise sparsity $vs$ and preserved proportion $ps$ after pruning. Our goal is to demonstrate that $R(vs, ps)$ is concave, implying the existence of a global optimum when jointly searching $vs$ and $ps$.

Let $W \in \mathbb{R}^{m \times n}$ be the original weight matrix with column-wise importance scores sorted in descending order. The vector pruning stage removes a proportion $vs$ of column vectors, preserving a fraction $(1 - vs)$. The following 2:4 pruning stage retains exactly $50\%$ of the weights in each preserved column. For a given target sparsity $ts$, the total preserved portion is constrained such that:

$$ps = \frac{2(ts - vs)}{1 - vs}, \quad \text{and thus,} \quad R(vs) = W(ts - vs).$$

This formulation is consistent with HiNM-style hierarchical pruning, where $ps$ is a function of $vs$ under fixed global sparsity.

Assume that the weight importance scores across columns follow a monotonically decreasing curve $W(c)$, where $c \in [0, C]$ is the column index after sorting. We can then express the remaining weight magnitude as:

$$R(vs) = W\big(c(vs)\big) \cdot (ts - vs),$$

where $c(vs) = C(1 - vs)$, and $C$ is the total number of column vectors.

Let us locally approximate $W(c)$ as a linear function: $W(c) = -ac + b$, where $a > 0$, $b > 0$. Substituting into $R(vs)$, we obtain:

$$R(vs) = (-aC(1 - vs) + b) \cdot (ts - vs) = (aCvs + b - aC) \cdot (ts - vs).$$

Taking the second derivative:

$$\frac{\mathrm{d}^2 R}{\mathrm{d}vs^2} = -2aC < 0.$$

Hence, $R(vs)$ is concave under this approximation. More generally, because $W(c)$ decays sublinearly in practice (due to long-tailed weight distribution), the composite function $R(vs) = W(c(vs)) \cdot (ts - vs)$ maintains concavity.

**Conclusion.** Since $R(vs)$ is a smooth, concave function on a compact domain $vs \in [0, ts]$, it admits a unique global maximum. This guarantees the existence of an optimal solution for selecting $vs$ (and indirectly $ps$), which can be efficiently searched. Therefore, the hybrid pruning target is formally a concave objective.

## C    FLEXHINM KERNEL

Figure C.1 illustrates the kernel design for a model pruned with FlexHiNM-GP. The weight tensor is separated into sparse and dense tiles, each assigned to a different CUDA stream: **Stream0** for sparse tiles and **Stream1** for dense tiles. It is crucial to note that different tiles use different vector indices. Initially (①), both the weight and input matrices are stored in global memory. These matrices are transferred to the shared memory of each Stream Multiprocessor (②③), and subsequently loaded into the registers of the processing cores (④⑤). To fully utilize GPU parallelism, multiple thread blocks across different SMs process tiles concurrently. Additionally, each thread block loads the relevant rows of the input matrix into shared memory, specifically the input row vectors that correspond to the column vector indices of the weight tile.

**Stream0: sparse tile execution** Each thread block (e.g. thread block 0 and 1) loads its corresponding weight tile and only the relevant input vectors into shared memory. These input vectors are selected

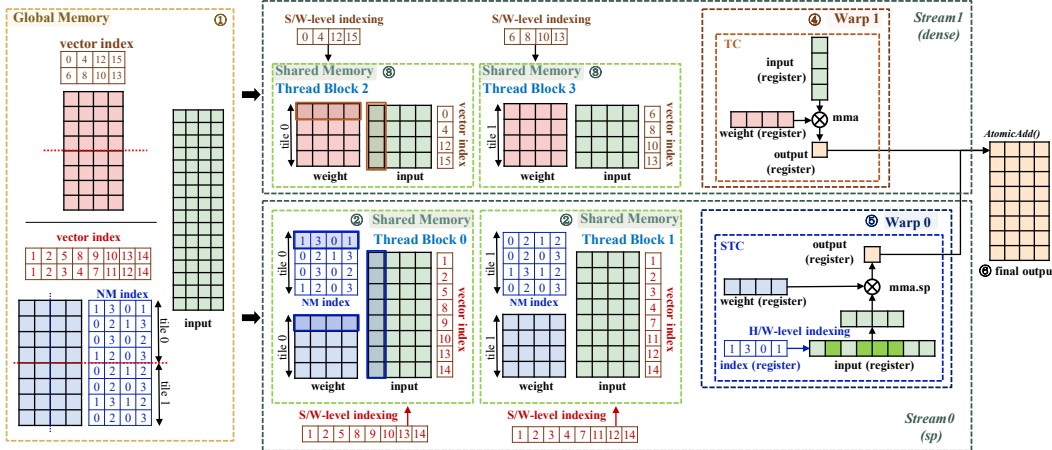

Figure C.1: FlexHiNM kernel

by analyzing the column vector indices retained in the sparse pattern (e.g. vector indices: 1, 2, 5, 8, 9, 10, 13, and 14 for thread block 0). During calculating (⑤), the STC engine loads each pruned weight row and its corresponding input row from registers, based on the precomputed N:M index (e.g. 1, 3 ; 0, 1). It performs sparse matrix multiplication using the `mma.sp` instruction in register space. This fine-grained activation-aware loading reduces unnecessary memory access, thereby maximizing throughput.

**Stream1: dense tile execution** In parallel, Stream1 handles dense tiles (③), such as thread blocks 2 and 3. These blocks load complete tiles from global memory using full vector indices (e.g. 0, 4, 12, 15 for thread block 2). Because no pruning is performed in these tiles, `mma` is applied using standard tensor cores (④), directly computing dense outputs.

**Final synchronization** Upon completion (④⑤), results from Stream0 and Stream1 are written directly into the global output matrix, where their partial contributions are accumulated using `atomicAdd` without requiring inter-stream synchronization.

## D  MONOTONICITY CONSTRAINTS

To ensure theoretical consistency and implementation safety of the gradual pruning schedule, we establish two necessary monotonicity conditions on the pruning variables. These properties serve as foundational guarantees for the correctness of our mask evolution mechanism.

**Case 1: Monotonicity of $vs$** : Let $vs$ denote the proportion of input vectors that have been entirely pruned. When the target sparsity $ts$ increases over time, $vs$ must not decrease. Allowing $vs$ to drop would imply that already-pruned vectors are reactivated, which violates pruning principles. Therefore, $vs$ must be non-decreasing with respect to $ts$:

$$\frac{\mathrm{d}vs}{\mathrm{d}ts} \geq 0.$$

**Case 2: Decreasing trend of remaining dense weights** : Let $N$ denote the as the number of total vectors in one weight tensor, and $M$ denote the number of dense weights remaining in the tensor. Given the analytical relationship:

$$M = N - N \cdot vs - 2N(ts - vs) = N(1 + vs - 2ts).$$

Then take $vs = (2ts - ps)/(2 - ps)$ into it and we get:

$$M = 2N \cdot \frac{1 - ps - ts + ps \cdot ts}{2 - ps},$$

we derive that $M$ is a decreasing function of $ts$, provided that $vs$ increases as $ts$ increases. Specifically:

$$\frac{dM}{dts} = \frac{ps - 1}{2 - ps} < 0 \quad \text{(for } ps < 1, \text{ always satisfied)},$$

and

$$\frac{\mathrm{d}^2 M}{\mathrm{d}ts^2} = \frac{1}{(2 - ps)^2} > 0.$$

According to the conclusion in **Case.1** that $vs$ must increase, $ps$ will also increase monotonicity.

This confirms that the number of remaining non-zero weights decreases smoothly with increasing sparsity target, avoiding oscillation or recovery of previously pruned weights. These two cases together justify the design of our pruning schedule and guarantee its progressive and irreversible nature.

## E  DETAILS OF THE LEARNABLE MASK ALGORITHM

To enable differentiable mask learning under the N:M sparsity constraint, we adopt the Hard Concrete distribution to parameterize binary masks in a continuous and learnable form.

**Step.1: Mask sampling** : To support gradient-based mask optimization, we update the binary mask every 20 epochs using a differentiable sampling procedure over grouped weights. We treat every row-wise group of four consecutive weights $\{w_1^t, w_2^t, w_3^t, w_4^t | t = 0, 1, \cdots, T\}$ as a unit $g$ to enforce the 2:4 sparsity constraint, which the set of all these $T$ groups is denoted by $\mathcal{G}$. Within each group, each element $w_i^t$ is associated with a learnable logit $\alpha_i$. Independently, a noise variable $\epsilon_i \sim \mathcal{U}(0, 1)$ is sampled for each element. These logits and sampled noise values are used to generate per-element soft masks $z_i$, which are later constrained at the group level to retain exactly two out of four elements.

---

**Algorithm 2** Sub-training for 20 epochs

---

**Require:** Weights $W$, Model $f(\cdot)$, inputs $X$, epochs $E$=20,
**Require:** Required hyper-params $(\tau_0, \tau_f)$, $(\gamma, \zeta)$, $\lambda_s$, $\lambda_h$
**Ensure:** Mask $M$
1: Establish 4-element groups $\mathcal{G}$ ▷ initial logits
2: **for all** $g \in \mathcal{G}$ **do**
3:     $\alpha_g \leftarrow \{0\}$
4: **end for**
5: Initialize the temperature $\tau \leftarrow \tau_f$
6: **for** epoch $= 1$ to $E$ **do**
7:     **for all** $g \in \mathcal{G}$ **do**
8:         Uniform sample: $\epsilon = \{\epsilon_i\} \sim \mathcal{U}(0, 1)$ for each $g$
9:         Generate the middle variable: $s \leftarrow \sigma\left((\log \epsilon - \log(1 - \epsilon) + \log \alpha)/\tau\right)$
10:         Strech and normalize $s$: $s \leftarrow s(\zeta - \gamma) + \gamma$ ; $z_{\text{soft}} \leftarrow \min(1, \max(0, s))$
11:     **end for**
12:     $\mathcal{Z} = \{z_{\text{soft}}\}; W \leftarrow \mathcal{Z} \odot W$
13:     $\mathcal{L}_{\text{task}} \leftarrow \text{loss}(f(W, X)))$
14:     $\mathcal{L}_{\text{sparse}} \leftarrow \lambda_s \cdot \text{mean}(\mathcal{Z})$
15:     $\mathcal{L}_{\text{hard}} \leftarrow \lambda_h \cdot \text{mean}\left(\left|\sum_{i=1}^{4} z_{\text{soft}}[g, i] - 2\right|\right)$
16:     $\mathcal{L} \leftarrow \mathcal{L}_{\text{task}} + \mathcal{L}_{\text{sparse}} + \mathcal{L}_{\text{hard}}$
17:     Back-propagate $\mathcal{L}$     $(W, \alpha) \leftarrow (W, \alpha) - \eta \nabla_{W, \alpha} \mathcal{L}$
18:     $\tau \leftarrow \tau_f - (\tau_f - \tau_0)/E \cdot \text{epoch}$
19: **end for**
20: **for all** $g \in \mathcal{G}$ **do**
21:     $z_{\text{hard}} \leftarrow \text{onehot}(\text{Top2}(z_{\text{soft}[g]}), 4)$
22: **end for**
23: Return $M \leftarrow \{z_{\text{hard}}\}$

---

**Step.2: Generate soft mask** $z_i$ : Given the sampled noise and logits from Step 1, we compute an intermediate variable $s_i$ that serves as a temperature-scaled activation.

$$s_i = \sigma\left(\frac{1}{\tau}(\log \epsilon_i - \log(1 - \epsilon_i) + \log \alpha_i)\right), \tag{6}$$

where $\sigma(\cdot)$ is the sigmoid function and $\tau \in [0.1, 1.0]$ is the annealed temperature. To enable mask learning over a broader range, $s_i$ is first stretched to the range $(\gamma, \zeta) = (-0.1, 1.1)$, and then clipped to $[0, 1]$ to obtain the final soft mask $z_i$:

$$\bar{s}_i = s_i \cdot (\zeta - \gamma) + \gamma, \tag{7}$$

$$z_i = \min(1, \max(0, \bar{s}_i)). \tag{8}$$

Then $z_i$ is applied to the corresponding weights through element-wise multiplication during forward propagation $\bar{w}_i = w_i \cdot z_i$ . Both the model weights and the sampling logits $\alpha_i$ are updated via backpropagation, guided by a composite loss:

$$L = L_{\text{task}} + L_{\text{sparse}} + L_{\text{hard}}$$
$$L_{\text{sparse}} = \lambda_s \cdot \text{mean}(z_i)$$
$$L_{\text{hard}} = \lambda_c \cdot \text{mean}\left(\left|\sum_{i=1}^{4} z_i - 2\right|\right) \tag{9}$$

where $L_{\text{task}}$ denotes the standard task loss, $L_{\text{sparse}}$ encourages overall sparsity by reducing the average mask value, and $L_{\text{hard}}$ enforces valid 2:4 sparsity within each four-element group $g$. In our implementation, the two regularization weights are set to $\lambda_s = 2 \times 10^{-4}$ and $\lambda_c = 1 \times 10^{-2}$. The mask variables $z_i \in [0, 1]$ remain continuous during training, so $L_{\text{sparse}}$ typically produces gradients of larger magnitude than $L_{\text{hard}}$. Setting $\lambda_c$ one order of magnitude larger than $\lambda_s$ balances the influence of the two terms: $\lambda_s$ softly drives the model toward a target sparsity level, while $\lambda_c$ penalizes deviations from the strict 2:4 pattern, ensuring that each group converges to a valid structured configuration before binarization.

**Step.3: Hardening and binary mask generation** : During training, the temperature $\tau$ is gradually annealed every 5 epochs, making the soft mask to hard mask.

$$z_i = \begin{cases} 1, & \text{if } z_i > 0.5 \\ 0, & \text{otherwise.} \end{cases} \tag{10}$$

Finally, at the end of each 20 epoch interval, we convert the soft mask into a hard binary mask by applying a 0.5 threshold to $z_i$, as defined in Equation 10. The procedure is repeated every 20 epochs, as summarized in Algorithm 2.

## F  GYRO-PERMUTATION

### F.1  OVERALL ALGORITHM

Gyro-Permutation is our pruning pipeline consisting of two sequential components: output channel permutation and input channel permutation. It first globally reorders output channels by importance to facilitate efficient vector removal, then locally adjusts input channels to enhance the effectiveness of N:M pruning.

HiNM pruning requires an effective permutation algorithm that accounts for both sparsity patterns across different channel dimensions. To design an effective permutation algorithm for HiNM, the following issues must be addressed:

1. Hierarchical Pruning Awareness: Since pruning is applied to both output and input channel dimensions in the HiNM method, an optimal channel permutation for one dimension may not be the best overall solution. For example, while an output channel permutation may cluster important elements into the same vector, subsequent N:M pruning might eliminate some of these preserved elements. This interdependence adds complexity to channel permutation in HiNM pruning.

2. Cross-Layer Consistency: Channel permutation must ensure that the channel orders are consistent between adjacent layers. Specifically, if the output channel order of the $i$-th layer is rearranged, the input channel order of the $(i + 1)$-th layer must be adjusted accordingly. Ensuring cross-layer consistency is straightforward when applying channel permutation to only one dimension. However, in HiNM, where both input and output channels are permuted in each layer, it becomes crucial to address the potential interference between layers during the permutation process.

3. Local Minima Problem: We observed that existing permutation techniques can easily fall into local minima. Addressing this challenge is essential for enhancing the performance of HiNM pruning.

For a given pre-trained model and target sparsity, a pruning method applies a mask $M$, adhering to specific structural constraints, to the weight matrix $W \in \mathbb{R}^{m \times n}$ in each layer, with the goal of maximizing the retention of important weight elements. A value of 0 in an element of $M$ indicates that the corresponding element in $W$ will be pruned. Channel permutation is an effective strategy to achieve this goal, as it rearranges channels to position important elements in the weight matrix in alignment with the structural constraints.

We assume that an importance score matrix $D \in \mathbb{R}^{m \times n}$ is given, where each element represents the importance score of the corresponding element in the pre-trained weight matrix. The pruning mask constraints, specifically the vector mask constraint $C_v$ and the 2:4 mask constraint $C_{2:4}$, must be taken into consideration. The constraint $C_v$ requires that all elements in a column vector must be either 1 or 0, while $C_{2:4}$ mandates that only two out of every four elements in a row vector can have a value of 1 in the mask. Additionally, the target sparsity constraint $C_s$ must be addressed. Since the sparsity achieved by 2:4 pruning is fixed at 50%, $C_s$ refers to the corresponding vector sparsity that satisfies the target sparsity. With these constraints in mind, the optimization problem for channel permutations in HiNM pruning can be defined as follows:

$$
\begin{aligned}
&\underset{\Lambda_O, \Lambda_I^0, ..., \Lambda_I^{T-1}}{\text{argmax}} && \| M \odot D[\Lambda_O; \Lambda_I] \| \\
&\text{s.t.} && M \text{ satisfies } C_v, C_{2:4}, \text{ and } C_s,
\end{aligned}
\tag{11}
$$

Here, $\Lambda_O$ represents the order of the output channels, while $\Lambda_I$ denotes the set of input channel orders for $T$ tiles, specifically $\Lambda_I^0, ..., \Lambda_I^{T-1}$, where $\Lambda_I^k$ is the channel order for the $k$-th tile. The notation $D[\Lambda_O; \Lambda_I]$ refers to the matrix $D$ rearranged according to the permutations $\Lambda_O$ and $\Lambda_I$. $\| \cdot \|$ represents the L1 norm.

The mask search problem encompasses a vast search space. To streamline the search for effective permutations, we decompose the problem into two sub-problems: output channel permutation and tile-wise input channel permutation. The output channel permutation aims to group less important elements within the same vector, thereby reducing the likelihood that important elements are pruned simply because they share a vector with unimportant ones.

However, it is not necessary to completely segregate important and unimportant elements into different column vectors, as elements retained after vector-wise pruning may still be removed during the subsequent N:M pruning. Instead, the focus is on creating a sufficient number of vectors containing only unimportant elements. The output channel permutation finds the vector pruning mask $M_v$ and can be formulated as follows using $\tilde{M}_v$, which denotes the inverted mask of $M_v$:

$$
\underset{\Lambda_O}{\text{argmin}} \left\| \tilde{M}_v \odot D[\Lambda_O] \right\| \quad \text{s.t. } M_v \text{ satisfies } C_v \text{ and } C_s.
\tag{12}
$$

After the output channel permutation, the importance score matrix $D$ is transformed into $D'$ by reordering channels and removing the pruned vectors ($D' = M_v \odot D[\Lambda_O]$). The next step is to optimize the arrangement of column-wise vectors within each tile to ensure that the weight distribution aligns with the desired uniform sparsity across row-wise vectors. It finds the 2:4 pruning mask $M_{2:4}$ and can be formulated as follows:

$$\underset{\Lambda_I^0, \ldots, \Lambda_I^{T-1}}{\arg\max} \; \|M_{2:4} \odot D'[\Lambda_I]\| \text{ s.t. } M_{2:4} \text{ satisfies } C_v. \tag{13}$$

## F.2 THE DETAILED ALGORITHM

Gyro-Permutation consists of three steps: output channel permutation, column vector pruning, and input channel permutation. The optimal permutation scheme will be obtained through a finite number of iterations, with each iteration process divided into three steps: sampling, clustering, and assignment, as illustrated in Figure 6. Gyro-Permutation can be applied iteratively to account for the interactions between output and input channel permutations.

**Step.1: Sampling.** For the output channel permutation, the weight matrix is first divided into several tiles, each containing column vectors within the same row (note that each tile corresponds to a tile in GPU execution). A fixed number of channels from all tiles are then selected and rearranged. In Figure 6, there are 9 rows of the original weight, while it is divided into 9/3=3 tiles, $P_0, P_1, P_2$ because the vector-size is 3. Then we assume that 2 samples are extracted from each tile.

The number of sampled channels plays a role analogous to the learning rate in optimization. Larger sample sizes promote global exploration and help escape poor local minima, whereas smaller sample sizes emphasize local refinement but risk premature convergence. To balance these behaviors, Gyro-Permutation adopts a sampling-and-clustering strategy that alternates between large-sample and small-sample steps. The sampling schedule follows a coarse-to-fine pattern (e.g., 8, 1, 4, 1, 2, 1, 1), where large-sample phases broaden the search space, and small-sample phases stabilize the configuration after each perturbation. This alternating update prevents both oscillation and early stagnation.

The effectiveness of the permutation is significantly influenced by the number of samples extracted from each tile, much like the role of learning rates in model training. Generally, extracting a larger number of samples helps avoid local minima but may make it harder to reach the absolute optimum. Conversely, extracting fewer samples can facilitate reaching the optimum but increases the risk of encountering local minima. In iterative Gyro-Permutation, the number of samples can be dynamically adjusted in each iteration, much like how learning rates are adapted during model training.

The input channel permutation for N:M sparsity is applied on a tile-wise basis, meaning each tile can have a unique permutation. The tiles used in the output channel permutation are further divided into sub-tiles, each consisting of $M$ column vectors, where $M$ is the basic vector size defined by N:M sparsity. Due to the limited size of these sub-tiles, only one column vector from each sub-tile is selected for permutation.

**Step.2: Clustering.** Clustering is used to group channels with similar importance scores, creating $T$ equally sized groups, where $T$ is the number of tiles.

After Sampling step, importance scores for each candidate channel are calculated as features input into the Clustering step. We use the Balanced K-means clustering algorithm as it effectively addresses the issue of uneven sample sizes, ensuring that the number of channels in each cluster is approximately equal. As shown in Figure 6, $s_0, s_1, s_2$ represent the sample groups after clustering.

**Step.3: Assignment.** During this phase, sample groups after clustering will be assigned into distinct tiles. The target tile for each group is determined by a cost function $C_{i,j}$, which represents the pruning cost when the $j$-th sample group is assigned to the $i$-th tile. This value is derived by implementing the target pruning. For example, in Figure 6, $C_{1,2}$ has a value of 0.2.

After calculating the cost for all combinations of tiles and samples, the Hungarian algorithm is used to find the combination that minimizes the total cost. These three steps will be repeatedly executed until the cost of a given iteration exceeds that of the previous one. In the example shown in Figure 6, the samples $s_0$, $s_1$, and $s_2$ are assigned to tiles $P_2$, $P_0$, and $P_1$, respectively.

**Overall** To facilitate the finite iterations, a list $J = j_0, j_1, \cdots, j_Y$ is given, wherein each element specifies the number of channels to be extracted during the sampling phase of each iteration. The overall permutation process is illustrated in Algorithm 3. By changing the direction of channel reordering, we combine functions `GyroPerm(weights,axis=output)` and

GyroPerm(weights,axis=input), which are applied respectively before the vector pruning and the N:M sparse pruning stages.

---

**Algorithm 3** Permutation Iteration Process

---

**Require:** Weight matrix $W$, iteration scheme list $J$, number of tiles $T$
**Ensure:** Updated channel assignments after permutation
1: Initialize partitions $P_0, P_1, \ldots, P_T$ derived from $W$ and $T$
2: **for** each number of extracted channels $j$ in $J$ **do**
3:     **Sampling:**
4:     **for** each $p$ in $P$ **do**
5:         Extract $j$ channels from $p$ to form sample groups
6:     **end for**
7:     **Clustering:**
8:     $s_0, s_1, \ldots, s_T \leftarrow$ BalancedKmeans(sample groups)
9:     Cost matrix $C_{i,j} \leftarrow$ CostFunction$(P_i, s_j)$
10:    **Assignment:**
11:    $current\_cost \leftarrow$ Hungarian$(C)$
12:    **if** $current\_cost \geq previous\_cost$ **then**
13:        **break**
14:    **else**
15:        $previous\_cost \leftarrow current\_cost$
16:    **end if**
17: **end for**

---

## G  SUPPLEMENTARY EXPERIMENTS

We evaluate `FlexHiNM-GP` on Deit-Small and Deit-Base under various sparsity levels, and compare it with `Unstructured`, `OVW`, `HiNM-V`, and `HiNM-GP` baselines.

Figure G.1 presents the results for the Deit-Small and Deit-Base models using vector-size of 128, with the baseline accuracy of 79.80% and 81.80%, respectively: `Unstructured` corresponds to element-wise pruning; `HiNM-GP` represents the HiNM pruning with Gyro-permutation; `FlexHiNM-GP` denotes the flexible Hierarchical pruning via region allocation and Gyro-permutation; `OVW` denotes the traditional out-vector-wise pruning, as a baseline since it can be regarded as a special case of our framework, representing a lower bound of accuracy for FlexHiNM; `HiNM-V` is a variant of our method that omits the Gyro-Permutation process, which is mainly equivalent to Venom, to isolate the effect of channel permutation.

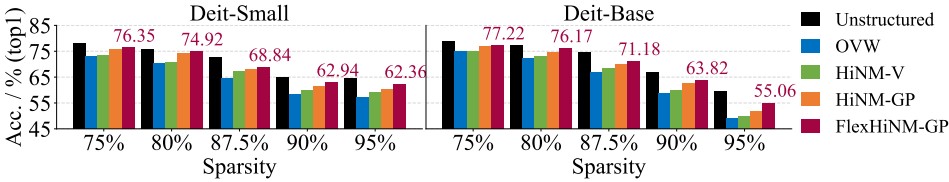

Figure G.1: Gradual pruning for Deit family (V=128)

As shown in Figure G.1 with vector-size of 128, `FlexHiNM-GP` outperforms `HiNM-GP` by 1.45% on Deit-Small and by 1.28% on Deit-Base at 90% sparsity,. The advantage becomes more pronounced under extreme sparsity: at 95%, `FlexHiNM-GP` surpasses `HiNM-GP` by 2.19% on Deit-Small and by 3.16% on Deit-Base. It demonstrates that `FlexHiNM-GP` is able to maintain higher accuracy under aggressive compression, indicating the effectiveness of its hybrid sparsity design and permutation-aware optimization.

Moreover, we evaluate `FlexHiNM-GP` on Bert-Base across three standard NLP benchmarks: QQP, SST-2, and SQuAD. Experiments are conducted under two total sparsity levels (75% and 87.5%) under the vector-size of 128, as shown in Figure G.2. `FlexHiNM-GP` consistently delivers competitive or superior performance compared to structured baselines across all settings.

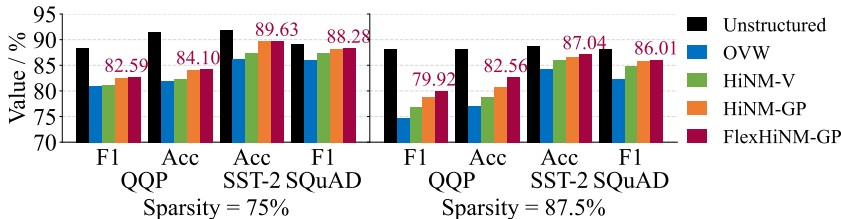

Figure G.2: Gradual pruning for Bert-Base (V=128)

Under 75% sparsity, `FlexHiNM-GP` achieves high task performance while maintaining structured constraints. When increasing vector size to 128, the model remains stable, achieving 88.28 on SQuAD and 89.63% on SST-2, again on par with `HiNM-GP` and outperforming `OVW` across the board. As sparsity increases to 87.5%, performance degradation is observed in all methods, yet `FlexHiNM-GP` maintains strong robustness. In the most challenging setting (87.5% sparsity with vector size 128), `FlexHiNM-GP` retains competitive performance: 86.01 of F1 score on SQuAD and 87.04% of accuracy on SST-2, exceeding `HiNM-GP` by 0.4 and 0.42 points, and significantly outperforming `OVW` by a large margin. The results highlight `FlexHiNM-GP`'s ability to preserve task performance under aggressive compression, showcasing the benefits of its permutation-guided design and differentiable masking in structured sparsity regimes.

