# OpenReview forum: "FlexHiNM-GP: Flexible Hierarchical Pruning via Region Allocation and Channel Permutation"
_ICLR.cc/2026/Conference — ICLR 2026 Poster_

### Official Review · Reviewer_mA3Q · 2025-10-29

**Soundness:** 3
**Presentation:** 3
**Contribution:** 3
**Rating:** 6
**Confidence:** 3

**Summary:**

This paper presented FlexHiNM, a framework to consider a hierarchical N:M sparsity (HiNM) using Gyro-permutation. Difference from HiNM that has 0:4 and 2:4 pruning, the proposed method considers 4:4 pruning that preserves entire vectors with high importance scores. Gyro-permutation based channel reordering optimizes the order of input and output channels so that important weights are well aligned with the n:M constraint. Hard concrete module enables differentiable mask learning under the N:M sparsity constraint. By applying DeiT, BERT, and LLaMA-2, the proposed method shows better performance than other previous methods such as HiNM and OVW (Outer-vector-wise Pruning)

**Strengths:**

- Compared with other methods dealing with N:M sparsity, this paper introduces a hierarchical structure of 0:4, 2:4, 4:4 pruning to provide variable sparsity. This ratio can be automatically adjusted based on the importance of each layer and channel to minimize accuracy loss.
- Gyro-permutation makes better aligned input and output channels with N:M sparsity.
- Hard concrete module shows better importance representation and stable convergence.
- By redesigning the GPU kernel to handle permutation operation, the gyro-permutation can be applied efficiently in real hardware environments.
- It shows better performance than HiNM when applying to DeiT, BERT, LLaMA2.

**Weaknesses:**

- The learning scheme in this paper has complex structure so that model training and implementation may be difficult. Due to various hyperparameters and boundary search (vs, ps), the training time should be increased.
- The additional memory and computational overhead are necessary for the process of permutation and learnable masks.
- Comparison with other methods considering N:M sparsity is necessary to show the effectiveness of the method, for example, Venom (Castro et al., 2023)
- There is no comparison of inference time or throughput, FLOPS.

**Questions:**

- Gyro-permutation for N:M sparsity is proposed in arxiv paper, Toward Efficient Permutation for Hierarchical N:M Sparsity on GPUs (2024). Why did not you cite and compare this paper?

---

> ### Author Response · Authors · 2025-11-21
>
> We thank Reviewer mA3Q for the careful reading and for identifying issues related to our working.
>
> # Q1: Relationship between variants
>
> Both VENOM and HiNM-V eliminate unimportant vectors and apply 2:4 pruning to the remaining ones; the key difference is that VENOM relies on input-channel grouping while pruning vectors, whereas HiNM-V does not, making our approach more flexible. We use HiNM-V as the baseline and extend it with GP and flexible region allocation, forming a direct progression in our framework, as illustrated in Figure B.1 in the Appendix.
>
> # Q2: Further consideration about trade-off
>
> We provide full wall-clock cost. Pruning BERT-Base on SQuAD1.1 to 75% sparsity takes 9.17 hours on 8×4090 GPUs, mostly from the Hard-Concrete sparse fine-tuning phase, which is the main computation of our method.
>
> The table summarizes the wall-clock cost of our pruning schedule. Each Stage corresponds to one step in the gradual sparsity progression (40→50→60→75%), with Recovery providing stabilization. BS records the vs/ps boundary search time, HC is the Hard-Concrete sparse fine-tuning, and Stage Time is their sum.
>
> | Stage |Target Sparsity(%) | Epoch | BS(h) | HC(h)| Stage Time(h) |
> |--------------------|----------------------|-------|----------------------|--------|-----------------|
> |**Total_Sparse_FT**|**75**|**100**|**1.38**|**7.78**|**9.17**|
> |│Recovery 0|30|5|0.00|0.32|0.32|
> |│Stage 1|40|20|0.45|1.82|2.27|
> |│Stage 2|50|20|0.35|1.63|1.98|
> |│Stage 3|60|20|0.30|1.48|1.78|
> |│Stage 4|75|20|0.28|1.28|1.57|
> |│Recovery 1|75|15|0.00|1.25|1.25|
> |**Total_FT**|**0**|**100**|**0.00**|**6.77**|**6.77**|
>
> The standard 100-epoch dense fine-tuning run already takes 6.7 hours (“Total_FT”) on the same hardware. The actual extra cost is limited to Boundary Search (1.38 hours) and GP (under 20 minutes per stage), making the overall overhead modest relative to typical training.
>
>
> # Q3: Further evaluation about CUDA kernel
>
> Our kernel does not incur uncoalesced global-memory accesses. The sparse weight format stores all nonzeros contiguously per warp, enabling fully coalesced weight loads. In addition, we reorder the input into an (out_dim, in_dim) layout, ensuring that each col_idx lookup accesses a long, contiguous segment of the input vector. This organization aligns naturally with cache-line boundaries and avoids scattered or irregular memory fetches.
>
> The revised paper now includes latency measurements. As expected, vector-wise pruning (VW) achieves the fastest execution because it loads only the input elements actually required for computation. HiNM, in contrast, incurs additional overhead since it loads the full input vector but uses only a subset of elements, leading to lower efficiency than VW. FlexHiNM processes both dense (4:4) vectors and 2:4-pruned vectors, so its performance naturally falls between VW and HiNM.
>
> |Method|**DeiT-Base**||| |**BERT-Base**|||
> |------|-------------|---|---|-|-------------|---|---|
> | |**75%*|**87.5%**|**90%**| |**75%**|**87.5%**|**90%**|
> |Dense|1.00|1.00|1.00| |1.00|1.00|1.00|
> |VW64|2.05|2.41|2.87| |2.16|2.55|3.23|
> |HiNM-GP|1.76|1.83|2.19| |1.86|2.21|2.83|
> |**FlexHiNM-GP**|**1.96**|**2.22**|**2.65**| |**2.09**|**2.46**|**3.05**|
>
> In our experiments on DeiT-Base and BERT-Base (batch size = 1, averaged over 30 post–warm-up runs on an RTX 4090), HiNM-GP achieves 1.7–2.8× speed-up over dense execution depending on sparsity, while FlexHiNM-GP further improves this to 1.9–3.05×, reaching performance comparable to VW(64) (where “64’’ indicates the tile-level vector-wise width).
>
>
> We hope these explanations effectively address the reviewer’s concerns.

---

### Official Review · Reviewer_LCAU · 2025-10-30

**Soundness:** 3
**Presentation:** 3
**Contribution:** 2
**Rating:** 6
**Confidence:** 4

**Summary:**

This paper addresses the lack of flexibility in N:M sparsity by proposing the FlexHiNM framework. The method partitions weights into three regions: dense (4:4), sparse (2:4), and pruned (0:4). To accommodate this partitioning, the authors design the Gyro-Permutation (GP) algorithm to reorder channels. The authors claim their method outperforms the compared structured pruning baselines on DeiT, BERT, and LLaMA-2 in terms of accuracy.

**Strengths:**

1. The motivation is clear and practically relevant. For hardware-supported N:M sparsity, the fixed 50% sparsity ratio is indeed a severe limitation in practice. The paper's direction in addressing this is correct.

2. The "three-region" partitioning (4:4, 2:4, 0:4) is a logical and incremental improvement over existing HiNM (2:4 and 0:4), providing finer-grained control for layers with different redundancies.

3. The paper conducts extensive experiments on accuracy metrics. The experimental results show that the proposed FlexHiNM-GP method does consistently surpass baselines like OVW and HiNM-V in accuracy on vision and language tasks, and it narrows the gap with unstructured pruning.

**Weaknesses:**

1. The paper is lack of practical inference latency evaluation. The paper's premise is hardware-friendliness and efficiency, yet the entire manuscript contains no data on end-to-end latency, throughput, or actual wall-clock speedup. The custom kernel proposed in the appendix is merely a theoretical design; its practical overhead from mixed execution and cross-stream synchronization is unknown. Lacking this core speed evidence, the paper's accuracy improvements are unconvincing, as they might be achieved at the cost of sacrificing inference performance.

2. The algorithmic proposal appears overly complex, like a hodgepodge of techniques. Its core innovation is limited to the three-region partitioning, which is a minor incremental improvement. Other parts of the framework, such as Hard Concrete mask learning (Algorithm 3) and gradual pruning, are off-the-shelf techniques in the field. The core Gyro-Permutation (GP) algorithm (Algorithm 2) feels like an over-designed product, seemingly designed as an extremely convoluted process (sampling + K-means + Hungarian matching) just to force the weight distribution to fit the three-region partitioning. This combination of boundary search, iterative GP, and gradual learning leads to extremely high pruning process overhead. The authors do not quantify this cost, making its practical feasibility questionable.

3. The ablation studies fail to sufficiently demonstrate the necessity of its core components. The most critical comparison is missing: a comparison between FlexHiNM (with only three-region partitioning) and FlexHiNM-GP (which adds Gyro-Permutation). We cannot ascertain whether the accuracy gain comes from the (relatively simple) three-region partitioning or from the (extremely complex) GP algorithm. If the improvement from the GP algorithm is negligible, its existence only serves to add unnecessary complexity.

**Questions:**

1. Why did you not provide any end-to-end inference latency or throughput data? This is the most critical metric for evaluating a hardware-friendly pruning method. Can you provide the actual speedup of FlexHiNM-GP versus HiNM-V at 75% and 87.5% sparsity, measured on an RTX 4090 using the kernel design from Appendix D.1?

2. Can you quantify the total computational overhead of the complete FlexHiNM-GP pruning pipeline (including boundary search, GP reordering, and mask learning)? For example, how many GPU-hours are required to prune the LLaMA-7B model once?

3. Can you add an ablation study comparing only FlexHiNM (without GP) versus FlexHiNM-GP? This is essential to justify the necessity of the complex Gyro-Permutation algorithm.

---

> ### Author Response · Authors · 2025-11-21
>
> We thank Reviewer LCAU for the careful reading and for identifying issues related to our working.
>
> # Q1: Inference latency evaluation
>
> Our kernel does not incur uncoalesced global-memory accesses. The sparse weight format stores all nonzeros contiguously per warp, enabling fully coalesced weight loads. In addition, we reorder the input into an (out_dim, in_dim) layout, ensuring that each col_idx lookup accesses a long, contiguous segment of the input vector. This organization aligns naturally with cache-line boundaries and avoids scattered or irregular memory fetches.
>
> The revised paper now includes latency measurements. As expected, vector-wise pruning (VW) achieves the fastest execution because it loads only the input elements actually required for computation. HiNM, in contrast, incurs additional overhead since it loads the full input vector but uses only a subset of elements, leading to lower efficiency than VW. FlexHiNM processes both dense (4:4) vectors and 2:4-pruned vectors, so its performance naturally falls between VW and HiNM.
>
> |Method|**DeiT-Base**||| |**BERT-Base**|||
> |------|-------------|---|---|-|-------------|---|---|
> | |**75%*|**87.5%**|**90%**| |**75%**|**87.5%**|**90%**|
> |Dense|1.00|1.00|1.00| |1.00|1.00|1.00|
> |VW64|2.05|2.41|2.87| |2.16|2.55|3.23|
> |HiNM-GP|1.76|1.83|2.19| |1.86|2.21|2.83|
> |**FlexHiNM-GP**|**1.96**|**2.22**|**2.65**| |**2.09**|**2.46**|**3.05**|
>
> In our experiments on DeiT-Base and BERT-Base (batch size = 1, averaged over 30 post–warm-up runs on an RTX 4090), HiNM-GP achieves 1.7–2.8× speed-up over dense execution depending on sparsity, while FlexHiNM-GP further improves this to 1.9–3.05×, reaching performance comparable to VW(64) (where “64’’ indicates the tile-level vector-wise width).
>
> # Q2: Further consideration about trade-off
>
> a) Each component targets a different limitation of HiNM: Boundary Search (BS) selects region-wise sparsity, Gyro-Permutation (GP) fixes channel-order misalignment, and Hard-Concrete optimizes the fine-grained 2:4 mask. Because these choices interact, iterative updates jointly adapt all configurations rather than treating them as separate steps.
>
> b) The three-region design is a substantive extension. Mixing 4:4, 2:4, and 0:4 regions lets each layer adapt to heterogeneous sensitivity, and the 4:4 region reduces unnecessary input loads, improving latency under N:M sparsity.
>
> c) Hard-Concrete here operates under an explicit N:M constraint, producing different masking behavior from unconstrained pruning. GP’s sampling–clustering process is also tailored for two-axis channel reordering, enabling effective exploration without falling into local minima.
>
> d) We provide full wall-clock cost. Pruning BERT-Base on SQuAD1.1 to 75% sparsity takes 9.17 hours on 8×4090 GPUs, mostly from the Hard-Concrete sparse fine-tuning phase, which is the main computation of our method.
>
> The table summarizes the wall-clock cost of our pruning schedule. Each Stage corresponds to one step in the gradual sparsity progression (40→50→60→75%), with Recovery providing stabilization. BS records the vs/ps boundary search time, HC is the Hard-Concrete sparse fine-tuning, and Stage Time is their sum.
>
> | Stage |Target Sparsity(%) | Epoch | BS(h) | HC(h)| Stage Time(h) |
> |--------------------|----------------------|-------|----------------------|--------|-----------------|
> |**Total_Sparse_FT**|**75**|**100**|**1.38**|**7.78**|**9.17**|
> |│Recovery 0|30|5|0.00|0.32|0.32|
> |│Stage 1|40|20|0.45|1.82|2.27|
> |│Stage 2|50|20|0.35|1.63|1.98|
> |│Stage 3|60|20|0.30|1.48|1.78|
> |│Stage 4|75|20|0.28|1.28|1.57|
> |│Recovery 1|75|15|0.00|1.25|1.25|
> |**Total_FT**|**0**|**100**|**0.00**|**6.77**|**6.77**|
>
> The standard 100-epoch dense fine-tuning run already takes 6.7 hours (“Total_FT”) on the same hardware. The actual extra cost is limited to Boundary Search (1.38 hours) and GP (under 20 minutes per stage), making the overall overhead modest relative to typical training.
>
> # Q3: Additional ablation study
>
> We provide an ablation study directly comparing FlexHiNM against FlexHiNM-GP. At 75% sparsity (vector size = 64), GP (Gyro-Permutation) consistently improves performance across all evaluated tasks on BERT-Base:
>
> | Method        | QQP F1 | QQP Acc | SST-2 Acc | SQuAD F1 |
> |---------------|--------|---------|-----------|----------|
> | FlexHiNM-GP   | 85.35  | 87.82   | 91.65     | 88.55    |
> | FlexHiNM      | 84.78  | 86.42   | 90.60     | 88.04    |
>
> GP provides stable gains across tasks in FlexHiNM, demonstrating that it corrects a real structural issue. These results show that three-region partitioning alone cannot resolve the channel misalignment that arises under structured N:M sparsity, particularly at higher sparsity levels. GP also yields significant accuracy improvements in HiNM, as shown in Fig. 7 and Table 1.
>
> We hope these explanations effectively address the reviewer’s concerns.

---

### Official Review · Reviewer_Cpf6 · 2025-11-01

**Soundness:** 3
**Presentation:** 2
**Contribution:** 3
**Rating:** 6
**Confidence:** 4

**Summary:**

This paper aims to address the effectiveness degradation caused by wide variance of important scores in N:M sparsity patterns. The authors propose a three-level hierarchical pruning framework that partitions model weights into dense, vector-pruned, and N:M-sparse regions, enabling balanced sparsity and accuracy. A boundary search algorithm is introduced to determine the optimal configuration. This paper also introduces Gyro-Permutation, a channel reordering method designed to enhance information aggregation and improve pruning performance. Combined with iterative pruning, the approach achieves further performance gains.
To mitigate the computational overhead introduced by permutation, a custom GPU kernel is developed to execute dense and sparse regions concurrently, ensuring hardware efficiency. The proposed framework is evaluated on both vision and language models, showing strong pruning performance.

Contributions

1. A three-level hierarchical pruning framework and a dynamical boundary searching algorithm.

2. The Gyro-Permutation algorithm that improves pruning efficiency through channel reordering.

3. A hardware-aware co-design, integrating a custom GPU kernel to offset the online computational cost of permutation.

**Strengths:**

**Originality.**
The paper provides original and meaningful improvements over prior structured pruning methods. It introduces an effective boundary search algorithm to adaptively determine three-level pruning ratios, a more robust channel permutation scheme (Gyro-Permutation) to enhance information concentration, and an efficient kernel implementation that avoids costly online computation.
Beyond these individual components, the work’s novelty lies in how it integrates algorithmic and hardware aspects into a unified co-design framework, achieving both flexibility and practical deployability.

**Quality.**
The technical quality of the paper is strong, particularly in the boundary search algorithm and the GPU kernel design. Both components are clearly formulated, well-motivated, and experimentally validated.

**Clarity.**
The motivation and high-level logic are clearly stated, helping readers understand how each module contributes to the framework.

**Significance.**
The work addresses a practically important challenge, improving the effectiveness of N:M sparsity under high variance, through a coherent combination of pruning and permutation.
The cross-domain applicability to both vision and language models increases its potential impact.

**Weaknesses:**

**Clarity of figures and presentation.**
Several figures (e.g. figure 6) are difficult to interpret and in some cases make the workflow harder to follow. A few textual descriptions are also inaccurate or ambiguous, which reduces readability and clarity (i.e. lambda in equation 3, what is the definition and what is the value in experiment).

**Insufficient detail on Gyro-Permutation.**
As a core component of the proposed framework, the Gyro-Permutation method is described too briefly in the main text and lacks sufficient algorithmic detail even in the appendix. Providing a clearer formulation and rationale would significantly strengthen the technical contribution.

**Questions:**

In Figure 1, why is the N:M range of R for the HiNM-P model bounded at 0.7?

The Figure 2 and Figure 3 are confusing to me. Which part in figure 2 corresponds to figure 3? There is a sorting logic in figure 3, but no corresponding illustration in figure 2.

The description of Figure 7 is confusing. It states that HiNM-GP is HiNM with gyro-permutation, but it is unclear which “HiNM” this refers to.
If HiNM-V is defined as FlexHiNM without gyro-permutation and is said to be “equivalent to Venom,” then the relationship among HiNM, Venom, HiNM-V, and HiNM-GP becomes ambiguous.
Does HiNM here correspond to Venom, or is it a different baseline?
If HiNM-GP = HiNM + gyro-permutation, and HiNM-V = Venom, then logically HiNM-GP should be equivalent to HiNM-V + gyro-permutation, which seems to describe FlexHiNM.
Please clarify the naming and hierarchical relationship among these variants.

What is the motivation behind gyro-permutation’s sampling and clustering?  In the appendix, it states that the number of samples is dynamically chosen without any detail. What is the method for choosing this number of samples hyperparameter? Also, there is no detail about how balanced K-means works.

For CUDA kernel, it seems when you move the input and weight from global memory to shared memory, you need to do indexing to select the corresponding partition. However, I wonder whether this indexing will cause an uncoalesced problem or not which may degrade the actual running speed? Since you have implemented the CUDA kernel, it would be more convincing if you provide experiment results on speed.

---

> ### Author Response · Authors · 2025-11-21
>
> We thank Reviewer Cpf6 for the careful reading and for identifying issues related to our working.
>
> # Q1: Clarifications on figures and formulas
>
> **a) Figure 1.**
> The 0.7 value in Fig.1(b) is only an illustrative example. It visualizes the reduced variance of importance scores after permutation. HiNM-P typically smooths the distribution and lowers the maximum score compared with the original HiNM case. The same trend appears in FlexHiNM-P (0.55 vs. 0.8). These values are not fixed bounds but represent the qualitative effect of permutation.
>
> **b) Figures 2 and 3.**
> Figure 3 summarizes the overall adaptive region-allocation process, whereas Figure 2 expands Steps 2–3: the upper block corresponds to vector pruning (Step 2) and the lower block to 2:4 pruning (Step 3). We added a short note in Section 3.2 to make this mapping explicit.
>
> **c) Figure 7.**
> HiNM in our paper refers to the general two-region template (2:4 / 0:4) in Fig. 1(a), not a specific variant. Section 6 introduces HiNM-V as the Venom-like instantiation, but without Venom’s input-channel grouping and fully integrated into our HiNM framework.
> Appendix Fig. B.1 clarifies these relationships.
>
> **d) Equation 3.**
> $ \lambda_s $ and $ \lambda_c $ control global sparsity and valid 2:4 patterns, respectively. The sparsity term $ L_{\text{sparse}} = \lambda_s \cdot \text{mean}(z_i) $ encourages low mask activation, while $ L_{\text{hard}} = \lambda_c \cdot \text{mean}(|\sum_{i=1}^{4} z_i - 2|) $ penalizes violations of the 2:4 constraint.
> Since $ z_i \in [0,1] $ during training, $ L_{\text{sparse}} $ produces stronger gradients; thus $ \lambda_c $ must be larger to balance the two signals. We use $ \lambda_s = 2 \times 10^{-4} $ and $ \lambda_c = 1 \times 10^{-2} $, a ratio of roughly 50, which yields stable sparsity and maintains valid 2:4 groups without affecting convergence. The revised paper includes this explanation in Section 3.2 and the detailed hyperparameter settings in the Appendix.
>
>
> # Q2: Further explanation of Gyro-Permutation
>
> GP uses a sampling-and-clustering strategy to explore cross-block permutations without exhaustive search. It alternates large and small sampling steps (e.g., 8→1→4→1→2→1→1) to enable global exploration followed by fine-grained refinement, avoiding both premature convergence and large oscillations.
> After sampling, balanced K-means groups channels into clusters of near-equal size, preventing high-importance channels from collapsing into the same group and ensuring well-structured sparsity. The Hungarian algorithm then selects the minimum-cost assignment. Together, these steps provide a controlled stochastic procedure that explores new permutations while maintaining structural balance and stable convergence.
>
>
> # Q3: Kernel evaluation
>
> Our kernel does not incur uncoalesced global-memory accesses. The sparse weight format stores all nonzeros contiguously per warp, enabling fully coalesced weight loads. In addition, we reorder the input into an (out_dim, in_dim) layout, ensuring that each col_idx lookup accesses a long, contiguous segment of the input vector. This organization aligns naturally with cache-line boundaries and avoids scattered or irregular memory fetches.
>
> The revised paper now includes latency measurements. As expected, vector-wise pruning (VW) achieves the fastest execution because it loads only the input elements actually required for computation. HiNM, in contrast, incurs additional overhead since it loads the full input vector but uses only a subset of elements, leading to lower efficiency than VW. FlexHiNM processes both dense (4:4) vectors and 2:4-pruned vectors, so its performance naturally falls between VW and HiNM.
>
>
> | Method         | **DeiT-Base** |        |        | | **BERT-Base** |        |        |
> |----------------|---------------|--------|--------|-|---------------|--------|--------|
> |                | **75%**       | **87.5%** | **90%** | | **75%**       | **87.5%** | **90%** |
> | Dense          | 1.00          | 1.00   | 1.00   | | 1.00          | 1.00   | 1.00   |
> | VW(64)         | 2.05          | 2.41   | 2.87   | | 2.16          | 2.55   | 3.23   |
> | HiNM-GP        | 1.76          | 1.83   | 2.19   | | 1.86          | 2.21   | 2.83   |
> | **FlexHiNM-GP** | **1.96**      | **2.22** | **2.65** | | **2.09**      | **2.46** | **3.05** |
>
> In our experiments on DeiT-Base and BERT-Base (batch size = 1, averaged over 30 post–warm-up runs on an RTX 4090), HiNM-GP achieves 1.7–2.8× speed-up over dense execution depending on sparsity, while FlexHiNM-GP further improves this to 1.9–3.05×, reaching performance comparable to VW(64) (where “64’’ indicates the tile-level vector-wise width).
>
> We hope these explanations effectively address the reviewer’s concerns.

---

### Meta-Review · Area_Chair_5uzx · 2025-12-23

**Summary:**

The paper initially received positive reviews, with scores of 6 from all three reviewers. After the authors submitted their rebuttal, the reviewers did not pose any further questions.

The area chair has evaluated both the reviewers' comments and the rebuttal. The chair believes that the concerns raised by the reviewers have been adequately addressed. Consequently, the area chair recommends that the paper be accepted.

**Reviewer Concerns:**

**Reviewer Cpf6** raised concerns about i) the clarity of figures and presentation, ii) the lack of detail on Gyro-Permutation, and iii) the lack of running speed. The rebuttal offers more detailed explanations, and the revised paper includes latency measurements.

**Reviewer LCAU**  raised concerns about i) the lack of practical inference latency evaluation, ii) the over-complicated design, and iii) insufficient ablation studies. The authors' rebuttal addressed the concerns by providing additional tables, including Inference latency measurement, the wall-clock cost of our pruning schedule, and an ablation study comparing FlexHiNM against FlexHiNM-GP.

**Reviewer mA3Q** raised concerns about i) the complexity of the design, ii) the additional memory and computational overhead, iii) the lack of comparisons with other methods, and iv) the absence of data on inference time or throughput. These concerns were consistent with those of other reviewers and were addressed in the rebuttal.

**Reviewer Scores:**

The three reviewers did not respond to the rebuttal before the OpenReview incident occurred.

Based on the review comments and the rebuttal, the area chair anticipates that the three reviewers will maintain their scores (all **6**).

---

### Decision · Program_Chairs · 2026-01-26

Accept (Poster)